# Sterile Silicone Ring Tourniquets in Limb Surgery: A Prospective Clinical Trial in Pediatric Patients Undergoing Orthopedic Surgery

**DOI:** 10.3390/jpm13060979

**Published:** 2023-06-10

**Authors:** Kunhyung Bae, Gisu Kim, Amaal M. Aldosari, Yeonji Gim, Yoon Hae Kwak

**Affiliations:** 1Department of Orthopedic Surgery, Hanyang University Hospital, Hanyang University College of Medicine, 222 Wangsimni-ro, Seongdong-gu, Seoul 04763, Republic of Korea; bae_k_h@naver.com; 2Department of Orthopedic Surgery, Asan Medical Center Children’s Hospital, University of Ulsan College of Medicine, 88, Olympic-ro, 43-gil, Songpa-gu, Seoul 05505, Republic of Korea; rlarl9744@naver.com (G.K.); dr_amaal@hotmail.com (A.M.A.); yeonji520@naver.com (Y.G.); 3Department of Orthopaedic Surgery, Al Noor Specialist Hospital, Makkah 24242, Saudi Arabia; 4Department of Orthopaedic Surgery, Severance Hospital, Yonsei University College of Medicine, 50-1 Yonsei-ro, Seodaemoon-gu, Seoul 03722, Republic of Korea

**Keywords:** sterile silicone ring tourniquet, pediatric orthopedics, extremity surgery, bleeding control

## Abstract

Sterile silicone ring tourniquets (SSRTs) reduce intraoperative bleeding and provide a wide surgical view. Moreover, they reduce the risk of contamination and are cheaper than conventional pneumatic tourniquets. Our study describes the perioperative outcomes of sterile silicone ring tourniquet placement in pediatric patients undergoing orthopedic surgery. We prospectively recruited 27 pediatric patients aged < 18 years who underwent 30 orthopedic surgeries between March and September 2021. Following complete surgical draping, all operations were initiated by placing SSRTs. We investigated the demographic and clinical characteristics of these patients, details of the tourniquet used, and intra- and postoperative outcomes of tourniquet placement. Owing to the narrowness of tourniquet bands and tourniquet placement at the proximal ends of the extremities, wide surgical fields were achieved, without limiting joint range of motion. Bleeding control was effective. Tourniquets were applied and removed rapidly and safely, regardless of limb circumference. None of the patients experienced postoperative pain, paresthesia, skin problems at the application site, surgical site infections, ischemic problems, or deep vein thrombosis. SSRTs effectively reduced intraoperative blood loss and facilitated wide operative fields in pediatric patients with various limb sizes. These tourniquets allow quick, safe, and effective orthopedic surgery for pediatric patients.

## 1. Introduction

Tourniquet use in orthopedic surgery enables bloodless operations, and facilitates the identification of important anatomical structures while also decreasing the anesthetic and operation time. However, the tourniquet itself is related to various complications such as skin and nerve injury, rhabdomyolysis, deep vein thrombosis, or compartment syndrome. Accordingly, many types of tourniquet systems have been invented and used by surgeon’s preferences.

Conventional pneumatic tourniquets have been shown to control blood flow and reperfusion during surgical procedures effectively; moreover, their reusability also makes them economical [1]. Despite these advantages, there is a demand for other types of tourniquets in pediatric patients undergoing orthopedic surgery. Pneumatic tourniquet cuffs are relatively wide enough to block blood flow. As children have relatively short limbs, the wide cuffs of pneumatic tourniquets cover greater areas in children than in adults. This can be a major obstacle, especially in proximal limb surgery, because it might block the sight of the surgeon’s field of view [2]. Additionally, the short limbs of infants and toddlers have a conical shape at the thigh, which often results in unintentional cuff sliding events and causes loss of arterial blood occlusion. Moreover, because limb size and circumference vary according to age, it is difficult to determine the adequate cuff size and amount of pressure to apply in pediatric patients. In addition, the skin and soft tissues are more delicate in children than in adults, increasing the probability of skin injury or chemical burns in the areas where the tourniquet was applied [3]. Furthermore, there is an issue of contamination of the tourniquet cuff; it can be a potential source of microbial colonization and may increase the surgical infection rate.

Owing to these drawbacks, Esmarch bandage tourniquets have been regarded as alternatives to conventional pneumatic tourniquets. They are also reusable, easy to sterilize, and allow blood exsanguination regardless of the limb circumference of the patients. However, Esmarch bandages still have wide cuffs, making it difficult to control the amount of pressure at the application site which can cause soft tissue damage [4].

Therefore, sterile silicone ring tourniquets have been suggested as good alternatives to pneumatic tourniquets. These tourniquets have only 2 cm wide cuffs, provide even pressure at compression sites, and can be applied in aseptic conditions [5]. Additionally, because it can be located at a more proximal site, we can achieve a wider surgical field without the risk of contamination. Although sterile silicone ring tourniquets have been frequently used and their outcomes have been reported in adult patients, only a few retrospective studies have evaluated their effects in pediatric patients [2,6].

This study aimed to investigate the effectiveness of sterile silicone ring tourniquets in pediatric patients undergoing orthopedic limb surgery. Both intraoperative and postoperative outcomes of sterile silicone ring tourniquet application and complications in pediatric patients undergoing orthopedic limb surgery were prospectively evaluated. We hypothesized that silicone ring tourniquets can work successfully in pediatric patients without perioperative complications.

## 2. Materials and Methods

### 2.1. Patient Selection

This study was approved by the Severance Hospital institutional review board (IRB No. 1-2020-0076). Patients who visited our pediatric orthopedic clinic between March 1st and September 30th, 2021 were prospectively recruited. Informed consent was obtained from all enrolled patients and their parents. Patients were included if they were (1) aged < 18 years and (2) scheduled to undergo upper or lower extremity in pediatric orthopedic surgery. Patients were excluded if the (1) expected tourniquet time was more than 2 h, (2) had poor skin condition where tourniquet would be applied, (3) were undergoing hip or shoulder joint surgery, (4) had unstable limb fractures, or (5) had musculoskeletal infections. Finally, a total of 27 patients (14 male, 13 female with 30 limbs) were included in the analysis.

### 2.2. Application of Sterile Silicone Ring Tourniquet in Limb Surgery

All surgeries in this study were performed by a single senior pediatric orthopedic surgeon (YHK). Sterile silicone ring tourniquets (Rapband^®^; Rapmedicare, Gyeonggi-do, Republic of Korea) (Figure 1) are designed for specific limb circumferences to provide preset pressure; unlike conventional pneumatic tourniquets, which allow the operator to control the applied pressure. They applied to provide different pressures with 4 sizes; small, medium, large, and extra-large sterile silicone ring tourniquets respectively providing pressures of 200 ± 20, 230 ± 40, 310 ± 40, and 320 ± 20 mmHg. The sizes were categorized by patient limb circumference, the small model was fit for 14 to 21 cm, the medium was 22 to 39 cm, the large was 40 to 54 cm, and the extra-large was more than 54 cm. Each tourniquet comprised a sterile silicone ring wrapped in a stockinet and two pulled straps. After complete aseptic draping, the most appropriately sized sterile silicone ring tourniquet was selected after measuring the limb circumference of the occlusion site with sterile tape measure. After an approximate ring tourniquet was selected, it was applied to the limb. The tourniquet was placed on the distal part of the limb and the two straps were pulled to the proximal part of the limb. The sterile silicone ring was unrolled to its final location at the proximal site with exsanguination of the remaining blood. The final width of the cuff after application was 2 to 3 cm. After the main surgical procedures were completed, the tourniquet was removed using a blade or pair of scissors.

### 2.3. Investigated Variables and Statistical Analysis

We recorded patient demographic characteristics and tourniquet information, including age, sex, diagnosis, surgical procedure, laterality, tourniquet application area, limb circumference, tourniquet size, and application time. Tourniquet outcomes were grouped as intraoperative and postoperative outcomes. The intraoperative outcomes included tourniquet application and removal times, changes in elbow or knee joint range of motion (ROM) before and after tourniquet application, adequate operative field visualization, and bleeding control evaluated by the number of gauze pads used. Postoperative outcomes included skin condition at the application site, surgical site infection, ischemic complications (compartment syndrome and distal neurovascular compromise), and deep vein thrombosis. For the patients aged > 5 years, pain and paresthesia at the tourniquet site were evaluated 24 h after surgery. The pain was evaluated by a numerical rating scale (NRS), a pain screening tool which rates the pain from 0 (no pain) to 10 (worst pain). Statistical analyses were performed using Microsoft Excel 2010 (Microsoft, Redmond, WA, USA).

## 3. Results

The demographic characteristics of the patients and tourniquet information are presented in Table 1. The mean patient age was 9.8 ± 5.3 years (range: 1 to 17 years) and the mean limb circumference at the application site was 35.9 ± 16.1 cm (range: 15 to 65 cm). Following the manufacturer’s guide, all four sizes of tourniquets were used for operation in this study. All included operations were completed within 2 h of tourniquet application, with a mean operation time of 36.5 ± 29.7 min (range: 5 to 110 min). The types of operations performed were various; fracture reductions, soft tissue surgeries, deformity correction, and implant removal.

The perioperative tourniquet outcomes are presented in Table 2. All tourniquets were applied and removed within 15 s in the operative field. Knee or elbow joint ROM was the same before and after the tourniquet application. Therefore, there were no postural limitations during surgery (Figure 2).

The thin width of the sterile ring tourniquet made it possible to easily expose lesions in the proximal limb, as it provided a sufficient surgical field (Figure 3). For the aspect of bleeding control, the need for intraoperative gauze usage was only observed in 3 surgeries, and the other 27 surgeries did not need gauze to stop bleeding.

The postoperative evaluation showed there was no soft tissue damage, such as skin necrosis, abrasion, and bullae in all patients (Figure 4). There was no surgical site infection during follow-up periods. None of the patients experienced major problems such as compartment syndrome, deep vein thrombosis, or distal neurovascular compromise due to ischemic damage to the tourniquet. All patients aged ≥5 years reported there was no pain (NRS score 0) and neurological deficits around the tourniquet ring site 24 h after surgery.

## 4. Discussion

The sterile silicone ring tourniquet is a single-use device that enables the exposure of a larger proximal area compared to conventional pneumatic compression tourniquets. We conducted a prospective study on the perioperative outcome of a sterile silicone ring tourniquet for limb surgery with a circumference of 15 cm to 65 cm for patients aged 1 to 17 years. It was easily applicable and removable in the operation field. The volume of the sterile silicone ring tourniquet was small and did not restrict joint movement during surgery. The postoperative evaluation showed no evidence of surgical site infections, skin problems, ischemic changes, or any abnormal symptoms when the tourniquet application was completed within 2 h. This study found that sterile silicone ring tourniquets were also effective and safe in pediatric patients with varying limb sizes and circumferences.

### 4.1. Sterile Silicone Ring Tourniquets for Pediatric Patients

Limb circumference increases as children grow until they reach skeletal maturity [7,8]. Therefore, it is essential to select appropriate tourniquet cuffs and pressure based on the circumference of individual limbs in pediatric patients undergoing orthopedic surgery. Because pneumatic tourniquet cuffs must be sterilized for intraoperative use, the cuff size must be determined at least several hours before surgery. In contrast, cuff sizes for silicone ring tourniquets can be determined based on limb circumference by tape measuring after surgical draping. Even for various limb circumferences, it is possible to use it immediately because there are sterilized single-use cuffs prepared. In addition, the nature of pediatric fractures, such as supracondylar and lateral condylar fractures, can lead to intraoperative changes from pre-planned surgical methods, including conversion from closed reduction to open reduction [9,10]. This conversion is difficult in the absence of a sterilized pneumatic tourniquet. In contrast, silicone ring tourniquets can always be applied, even during unplanned alterations of the surgical methods; moreover, both application and removal can be completed within 15 s. In pediatric trauma surgery where a possibility of surgical method conversion may be required, the silicone ring tourniquet presents several advantages over conventional pneumatic tourniquet.

### 4.2. Intraoperative Outcomes of Sterile Silicone Ring Tourniquet-Surgical Fields

Sufficient operative fields were secured in all 30 limbs analyzed in this study. Especially in the proximal site of upper arm or thigh operations, a silicone ring tourniquet enables the operator a wide view (Figure 2). The traditional pneumatic cuff was too wide to be used in upper arm or thigh operation, however, the proximal limb length exposed by ring-type tourniquets is longer than that exposed by conventional pneumatic tourniquets [11,12]. This is because pneumatic cuffs are 8 to 14 cm wide when added to surgical drapes, making it difficult to expose the proximal surgical site in young children with short limb lengths. In contrast, silicone ring tourniquets provide better limb exposure because the final cuff width is approximately 2 cm. This has been a great advantage in operations that require maximal exposure of the upper thigh or arm, such as soft tissue tumor removal or proximal limb fixation surgery [13]. In addition, the application of silicone ring tourniquets did not decrease joint ROM. In obese patients, the thickness of inflated pneumatic cuffs makes it difficult to obtain full ROM during surgery (Figure 3b). In contrast, because they are narrower, silicone ring tourniquets do not alter joint ROM even after application. These tourniquets are not restricted by changes in posture, thus allowing for an easier surgical approach. Moreover, bleeding control with silicone ring tourniquets was similar to that of conventional tourniquets in adults undergoing orthopedic surgery [14]. In this study, most operations were bloodless and gauze was generally not required. Only three cases required gauze for bleeding control, two cases one piece of gauze, and one case two pieces of gauze. Therefore, this type of tourniquet also effectively minimizes blood loss in pediatric orthopedic surgery. In addition, the application and removal time of sterile silicone ring tourniquet is 7.5 s and 5.4 s, respectively. It is a fast and easy way because one just needs to apply the tourniquet by rolling it from the distal to the proximal part and remove it to cut the tourniquet with a scalpel. In contrast, conventional tourniquet application is much more complex. The application involves tightening the cuff for a snug fit by pulling the straps and fasteners around the limb in opposite directions.

### 4.3. Postoperative Outcomes of Sterile Silicone Ring Tourniquet

None of the patients in the present study experienced skin problems such as bullae, necrosis, hematoma, contusion, or burn wounds at the tourniquet application site. Because children have softer and more fragile soft tissue than adults, tourniquets may be harmful postoperatively [2]. There are several reports of burns, abrasions, or hematomas at the application site with the pneumatic tourniquet, however, proper application of skin protection can protect the soft tissue from damage [3,15]. In this study, none of the patients experienced skin or soft tissue problems even 24 h postoperatively. Especially, there were 4 patients who were ≤3 years old who had more delicate soft tissues than older children. Therefore, even in toddlers, the silicone ring tourniquet would not cause soft tissue damage if used within proper operation time.

In addition, none of the patients experienced surgical site infection. Although the reuse of pneumatic tourniquets is economical, they can be a source of infection even after ethylene oxide sterilization [16]. Such equipment has the potential to colonize bacteria, increasing the possibility of transmitting pathogens to patients through the operating room. These surgical site infections increase the medical burden and consume medical resources, but also have harmful effects on the patients by repetitive blood tests, antibiotic usage, increasing length of hospitalization, and the possibility of reoperation [17]. There is one study reported that bacterial contamination in two-thirds of orthopedic surgical tourniquets with normal flora, such as coagulase-negative *Staphylococcus* spp., Staphylococcus aureus, suggesting that bacteria may be transferred between operated patients [18]. In contrast, silicone ring tourniquets are both sterile and disposable, thereby reducing the risk of surgical site infections [18].

Tourniquet use has also been associated with ischemic complications and deep vein thrombosis. High pressure and prolonged obstruction of arterial blood flow induce ischemic changes in the limbs. Moreover, ischemic tissue perfusion after blood circulation resumes can lead to secondary injury, including compartment syndrome and distal neurovascular problems, with the remaining blood possibly causing deep vein thrombosis [4,19]. Ischemic soft tissue damage did not differ between silicone rings and pneumatic tourniquets; therefore, customary tourniquet time within 2 h is safe for both types of tourniquets [20]. In addition, silicone ring tourniquets effectively minimize residual blood in the extremities by compressing the limb while unrolling it from the distal end to the proximal site. This procedure provides effective exsanguination and may significantly reduce deep vein thrombosis. Tourniquet pain originated from both compression effects of anatomical structures and ischemic changes; however, which etiologies have a main role is debatable. None of the patients in the present study experienced pain or sensory problems at the tourniquet site within 24 h postoperatively. According to previous studies, the incidence of pain and paresthesia among patients who underwent silicone ring tourniquet application was comparable to or lower than the incidence of these complications among patients who underwent pneumatic tourniquet application; its incidence is 1/50,000 and 1/5000, respectively [12,21,22]. Lee et al. recently reported in adult studies that 13.3% of patients felt higher pain in the pneumatic tourniquet, 76.7% of patients felt the same, and 10% of patients experienced higher pain in the silicon ring tourniquet-applied lower extremity [23]. However, pain associated with the use of tourniquets was first studied in 1952 and a number of mechanisms have been proposed as the cause. The exact etiology is unclear, but it is thought to be due to a cutaneous neural mechanism. The incidence of tourniquet pain was directly related to the duration of tourniquet use and was higher in cases with regional anesthesia [24]. The most important point was neurological compromise occurred in the silicone ring tourniquet with over 120 min of application time, over the recommended time. You should adhere to the principle of tourniquet use, including the time of application. There was one study that compared the pain score of pneumatic compression and silicone ring tourniquet in adult patients who underwent local anesthesia, the results of pain score and paresthesia were significantly lower in the silicone ring tourniquet group [23]. Moreover, considering that continuously publishing results show that the narrow silicone ring tourniquet causes less nerve damage, neurological problems are also safe for children who use the silicone ring tourniquet [24].

The present study has several limitations. First, it was a prospective clinical trial and not a comparative study. The target patient population was heterogeneous, with patients undergoing different parts of limbs with kinds of surgery. However, the strength of this study is its diversity of patients group of various ages and limb sizes, because it implies that silicone ring tourniquet has versatility in the pediatric orthopedic field. Even though these characteristics are in pediatric studies, further study should be considered for comparison with current widely used pneumatic tourniquets or Esmarch bandages. Second, this study reported the short-term results of sterile silicone ring tourniquet application. However, most complications associated with tourniquet application appear within a short period of time, suggesting that a short follow-up period may provide significant results; nevertheless, long-term follow-up results are warranted. Finally, the study had a limited sample size, with only 30 cases among 27 patients included. Because silicone ring tourniquets were developed for use in adults, only a few studies have evaluated these tourniquets in pediatric patients. Therefore, future comparative trials with a larger number of patients are warranted.

## 5. Conclusions

Sterile silicone ring tourniquets have easy and fast applications compared to preset pressure models. As opposed to concerns about pressure-focused narrow tourniquets, this study showed no pain, nerve symptoms, or skin problems in vulnerable pediatric patients. In addition, unlike conventional pneumatic tourniquets or Esmarch bandages, SSRTs apply uniform pressure to the limb during the application, reducing the risk of deep vein thrombosis. These tourniquets provide a sufficient surgical field in pediatric patients undergoing orthopedic surgery on their extremities because they are located at a more proximal site on the extremities with narrow cuff width. Moreover, their application within 2 h can ensure successful bleeding control without soft tissue complications, neurovascular compression, or surgical site infection.

## Figures and Tables

**Figure 1 jpm-13-00979-f001:**
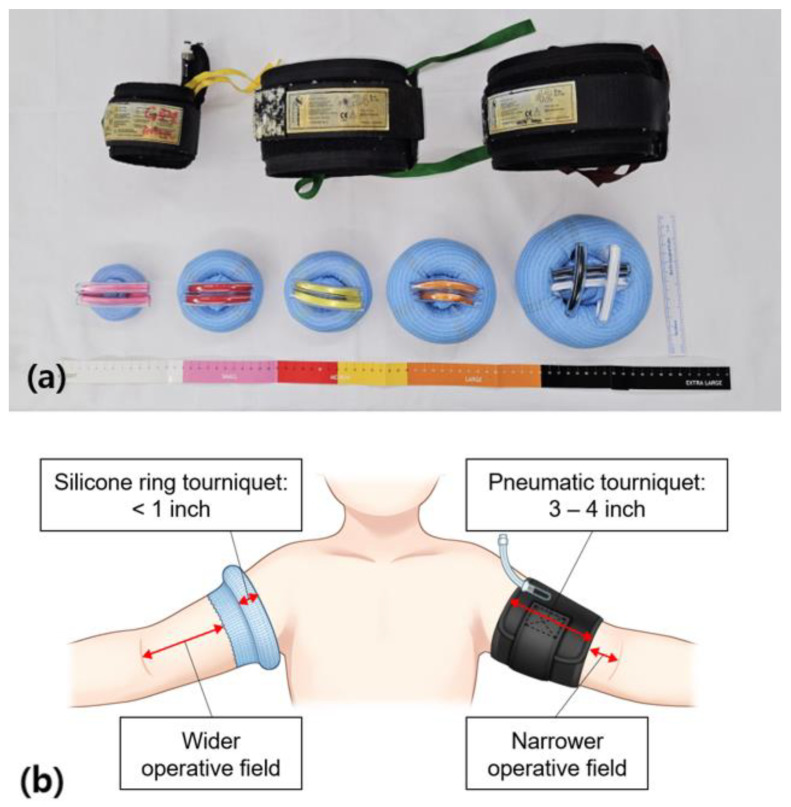
(**a**) Sterile silicone ring tourniquet and conventional pneumatic tourniquet, and (**b**) detailed illustration of applied states.

**Figure 2 jpm-13-00979-f002:**
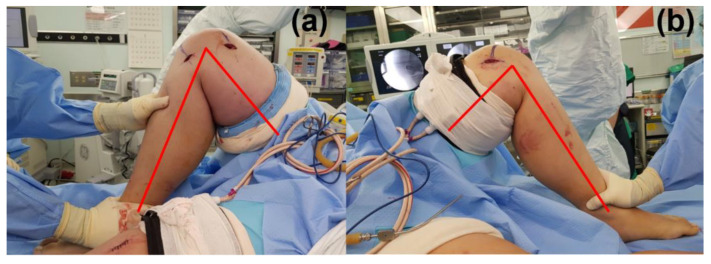
Knee joint range of motion (ROM) between sterile silicone ring and pneumatic tourniquets. Comparison of knee joint range of motion (ROM) between sterile silicone ring tourniquet in the right lower leg (**a**) and pneumatic compression tourniquet (**b**) in the left leg for a 12-year-old male patient with both side hemiepiphysiodesis. After the sterile silicone ring tourniquet application, the right knee joint still had full flexion with a wide operative field; however, the left knee joint showed limited flexion due to the inflated pneumatic cuff and narrow operative field.

**Figure 3 jpm-13-00979-f003:**
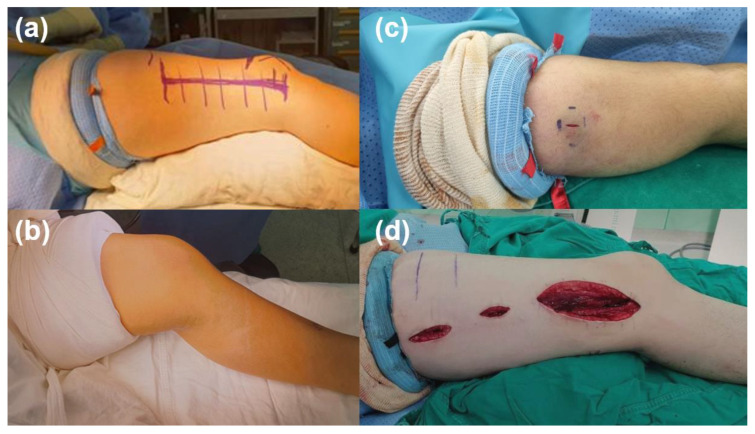
Size of surgical fields between sterile silicone ring and pneumatic tourniquets. Comparison of surgical fields for (**a**) a sterile silicone ring tourniquet and (**b**) a conventional pneumatic tourniquet. Sterile silicone ring tourniquet cuff is much narrower than conventional pneumatic tourniquet cuff (2 to 3 cm vs. 8 to 16 cm, which allows for better surgical site exposure for proximal thigh lesions. Surgical field after application of a sterile silicone ring tourniquet in (**c**) a 10-year-old girl with pilomatricoma excision at the upper right arm and (**d**) a 16-year-old boy with open reduction and internal fixation of a femur shaft fracture. Both patients required maximal exposure of the proximal limb, making it mandatory to use sterile silicone ring tourniquets rather than pneumatic tourniquets.

**Figure 4 jpm-13-00979-f004:**
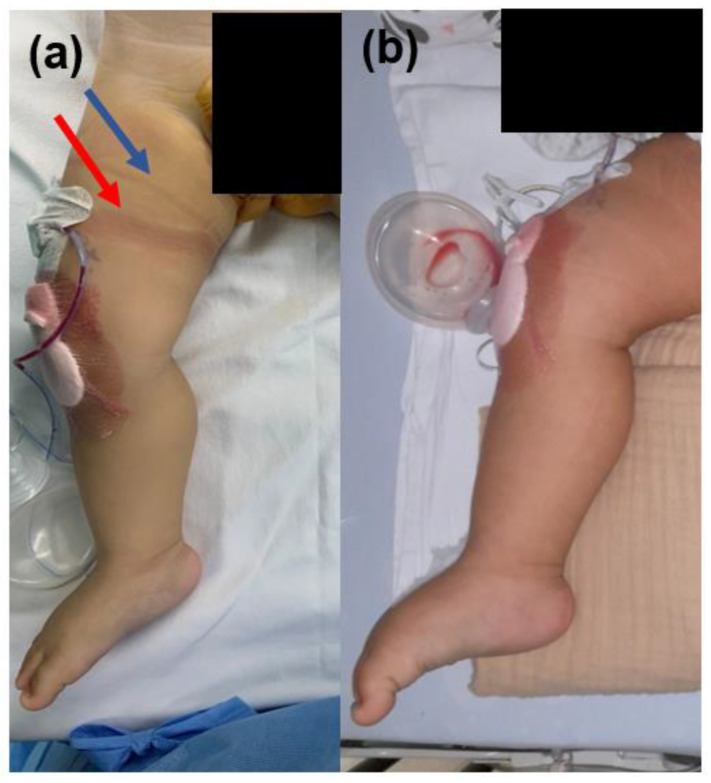
The reversible compression effect of the sterile silicone ring on the skin. (**a**) Compressed skin mark (red arrow) immediately after sterile silicone ring tourniquet removal. However, there was no bullae, blister, or hematoma. Blue arrow shows the marks on the skin after U-drape (**b**) Compressed skin lesion by silicone ring returned to a normal status 8 h after silicone ring tourniquet removal.

**Table 1 jpm-13-00979-t001:** Demographic and clinical characteristics of the patients and information of applied silicone ring tourniquets in this study.

Patient No.	Case no.	Age (year)	Sex	Diagnosis	Operation Procedure	Laterality	Application Area	Circumference ^1^ (cm)	Time (min)	Size
1	1	3	F	Congenital trigger thumb	A1 pulley release	R	Upper arm	17	11	S
2	2	4	M	Congenital trigger thumb	A1 pulley release	R	Upper arm	20	10	S
3	3	10	F	Pilomatrichoma	Mass excision	L	Upper arm	27	19	M
4	4	10	F	Ganglion cyst	Mass excision	R	Thigh	45	47	L
5	5	17	F	Lower leg deformity due to neonatal sepsis	Plate change	R	Thigh	50	73	L
	6			Lower leg deformity due to neonatal sepsis	Plate change	L	Thigh	48	51	L
6	7	12	F	Talocalcaneal coalition	Coalition resection	R	Thigh	49	59	L
7	8	15	F	Jones fracture	ORIF by screw	R	Thigh	58	31	XL
8	9	7	M	Both forearm fracture	CRIF with flexible elastic nail	L	Upper arm	23	21	M
9	10	14	M	Distal femur hemiepiphysiodesis status	Implant removal	L	Thigh	44	9	L
10	11	9	M	Femur shaft fracture fixation status	Implant removal	L	Thigh	37	48	M
11	12	14	F	Accessory navicular bone	Accessory bone resection	R	Thigh	47	12	L
12	13	17	F	Distal tibia fracture fixation status	Implant removal	L	Thigh	50	32	L
13	14	11	M	Trevor’s disease	Mass excision	R	Upper arm	23	29	M
14	15	16	F	4th toe epidermoid cyst	Mass excision	R	Thigh	60	26	X
15	16	4	M	Congenital trigger thumb	A1 pulley release	R	Upper arm	18	8	S
	17			Congenital trigger thumb	A1 pulley release	L	Upper arm	19	7	S
16	18	3	M	Congenital trigger thumb	A1 pulley release	L	Upper arm	18	7	S
17	19	1	M	Congenital trigger thumb	A1 pulley release	L	Upper arm	15	10	S
18	20	5	M	Congenital trigger thumb	A1 pulley release	L	Upper arm	23	12	M
19	21	5	M	Lateral condylar fracture fixation status	Implant removal	L	Upper arm	24	90	M
20	22	16	M	Distal femur fracture fixation status	Implant removal	L	Thigh	65	110	XL
21	23	12	M	Distal femur hemiepiphysiodesis status due to idiopathic genu valgum	Implant removal	R	Thigh	46	40	L
	24			Distal femur hemiepiphysiodesis status due to idiopathic genu valgum	Implant removal	L	Thigh	45	35	L
22	25	1	M	Hand preaxial polydactyly	Extra digit excision	R	Upper arm	15	80	S
23	26	12	F	Revisional Achilles tendon Z plasty	CMT with Achilles tightness	L	Thigh	43	85	L
24	27	4	F	Congenital trigger thumb	A1 pulley release	R	Thigh	17	5	S
25	28	12	M	Fifth finger proximal phalanx malunion	Deformity correction by pinning	R	Thigh	23	80	M
26	29	17	F	Lateral malleolar fracture fixation status	Implant removal	L	Thigh	47	35	L
27	30	14	F	Achilles tightness	Achilles Tendon lengthening	R	Thigh	60	12	XL

no, number; cm, centimeter; min, minute; M, male; F, female; R, right; L, left; S, small; M, medium; L, large; XL, extra-large; ORIF, open reduction and internal fixation; CRIF, close reduction and internal fixation; CMT, Charcot-Marie-Tooth disease. ^1^ Circumference at the tourniquet application area.

**Table 2 jpm-13-00979-t002:** Intraoperative and postoperative outcomes of sterile silicone ring tourniquets.

Tourniquet Outcome Parameters	Results
Intraoperative Outcome	
Tourniquet application time (s)	Mean: 7.5 ± 2.8 (range: 4 to 15)
Tourniquet removal time (s)	Mean: 5.4 ± 1.4 (range: 4 to 10)
Joint ROM between pre- and post-tourniquet application ^1^	All cases showed the same joint ROM after the tourniquet application
	All cases had a sufficient operative field for surgery even after applying a tourniquet
Operative field visualization	Gauze not used to control bleeding in 27 cases
Bleeding control (by gauze counting)	Two cases used one piece of gauze—ORIF by screw for Jones fracture, mass excision for Trevor’s disease
	One case used two pieces of gauze—Plate change for deformity due to neonatal sepsis
Postoperative outcome	
Skin problem at the tourniquet application site	None of these cases experienced skin problems, including bullae, necrosis, hematoma, contusion, or burn
Surgical site infection	None of these cases experienced surgical site infection after follow-up periods
Ischemic complications	None of these cases showed ischemic complications, including compartment syndrome or neuromuscular compromise
Deep vein thrombosis	None of these cases showed deep vein thrombosis
Pain at tourniquet application site ^2^	All 18 patients aged >5 years reported an NRS score of 0
Abnormal sensory change at tourniquet application site ^2^	None of the 18 patients aged >5 years experienced abnormal sensory changes

sec, second; ROM, range of motion; ORIF, open reduction, and internal fixation; NRS, numerical rating scale. ^1^ The elbow joint was evaluated for upper extremity surgeries, and the knee joint was evaluated for lower extremity surgeries. ^2^ Evaluated 24 h after surgery in patients aged ≥5 years.

## Data Availability

The datasets used and analyzed during the current study are available from the corresponding author on reasonable request.

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
