# Peer review of "Sterile Silicone Ring Tourniquets in Limb Surgery: A Prospective Clinical Trial in Pediatric Patients Undergoing Orthopedic Surgery"

_jpm, 2023, doi:10.3390/jpm13060979_

Round 1

Reviewer 1 Report

Dear authors,

Thank you for your valuable report. I have checked the manuscript. Comment below.

1. You guys should show the SSRTs graphically. If possible, I think it would be better to compare it with the conventional type and show it in a diagram. If you wrap it around your limbs, you will not be able to distinguish it from the conventional type.

2. How did you determine the tourniquet pressure? Is it the same as the conventional method?

3. In the "Consideration", it is written that it takes 15 seconds to put on and take off, but is there any reason why this is easier to put on and take off than the conventional method? The diagram shown shows a bandage over the hemostat. do you need this?

4. I think it's great that there is no pain after 24 hours. However, it would be useful for many researchers if there was information that neurological symptoms were likely to occur depending on the time, location, and size of bleeding. I think that it will be a better report if you describe the consideration with the conventional method in your own experiment based on scientific grounds.

Author Response

The responses to the reviewers’ comments and changes are presented in blue letters

Dear authors,

Thank you for your valuable report. I have checked the manuscript. Comment below.

  1. You guys should show the SSRTs graphically. If possible, I think it would be better to compare it with the conventional type and show it in a diagram. If you wrap it around your limbs, you will not be able to distinguish it from the conventional type.

Author’s answer:

 Thank you for your thoughtful reviews. We fully agree with your suggestions and added the comparison diagram between sterile silicone ring tourniquets (SSRTs) and conventional pneumatic tourniquets as Figure 1 (line 106). In this figure, we showed each tourniquet in its unapplied states, with its characteristics in place (a), and its applied states were depicted with illustrations (b).

Revisions:

1.Figure 1

  1. Line 106) Figure 1 legend: ‘(a) Sterile silicone ring tourniquet and conventional pneumatic tourniquet, and (b) detailed illustration of applied states’
  2. How did you determine the tourniquet pressure? Is it the same as the conventional method?

Author’s answer:

 We appreciate your valuable comment about determining tourniquet pressure. The tourniquet pressure decision of SSRTs is quite different from conventional pneumatic compression tourniquets. SSRTs are designed for a specific range of limb circumferences and maximum systolic blood pressure in four sizes. After aseptic draping, the most appropriate size sterile silicone ring tourniquet was selected after measuring the limb circumference at the occlusion site with a sterile tape measure. This is different from a traditional pneumatic tourniquet where the operator can directly adjust the pressure of the tourniquet. The reason why pressure is higher than conventional wide pneumatic tourniquet is that SSRTs requires a higher skin pressure to achieve sufficient tissue pressure to occlude the artery with less skin injury, tourniquet pain and tourniquet-induced nerve damage. [ref: Weatherholt, A.M.; Vanwye, W.R.; Lohmann, J.; Owens, J.G. The effect of cuff width for determining limb occlusion pressure: a comparison of blood flow restriction devices 2019, 12(3), 136-143.]

 We added detailed content about pressure of SSRTs in the manuscript (Line 87).

Revision:

  1. Line 87) ‘Sterile silicone ring tourniquets (Rapband; Rabmedicare, Gyonggi-do, Republic of Korea) (Figure 1) are designed for specific limb circumferences to provide preset pressure; unlike conventional pneumatic tourniquets, which allow the operator to control the applied pressure. They applied to provide different pressures with 4 sizes; small, medium, large, and extra-large sterile silicone ring tourniquets respectively providing pressures of 200 ± 20, 230 ± 40, 310 ± 40, and 320 ± 20 mmHg.’
  2. In the "Consideration", it is written that it takes 15 seconds to put on and take off, but is there any reason why this is easier to put on and take off than the conventional method? The diagram shown shows a bandage over the hemostat. do you need this?

Authors reply:

 Thank you for pointing this out. The reason SSRTs take several seconds to put on is that we simply roll SSRTs up like an anti-embolism stocking. To remove the SSRTs, we simply cut the SSRTs with a blade in seconds. On the other hand, applying a pneumatic tourniquet involves a more complicated method; tightening the cuff for a snug fit by pulling the cuffs and fasteners in opposite directions around the limb and then engaging the fasteners. Elastic or Esmarch bandages were then used to squeeze blood in the extremities;

In addition, a bandage of a SSRTs in the figure is initial drape with stockinet and a bandage in a pneumatic tourniquet is for prevention of blow out during the procedure due to Velcro weakening by multiple reused pneumatic tourniquets.

To follow your suggestion, we added an additional explanation on line 231 comparing SSRTs and pneumatic tourniquets in terms of application and removal time.

Revision:

  1. Line 231) ‘In addition, application and removal time of sterile silicone ring tourniquet is 7.5 seconds and 5.4 seconds, respectively. It is fast and easy way because you just need apply tourniquet by rolling it from distal to proximal part and remove to cut the tourniquet with scalpel. In contrast, conventional tourniquet application is much complex. Application involves tightening the cuff for a snug fit by pulling the straps and fasteners around the limb in opposite directions’
  1. I think it's great that there is no pain after 24 hours. However, it would be useful for many researchers if there was information that neurological symptoms were likely to occur depending on the time, location, and size of bleeding. I think that it will be a better report if you describe the consideration with the conventional method in your own experiment based on scientific grounds.

Authors reply:

 We totally agree with your comments. Randomized controlled study is much informative for comparison. However, when we designed this prospective study, we used Esmarch bandages for pediatric patients for several reasons. First, our hospital did not have various sized pneumatic tourniquets for pediatrics. Second, it was difficult to apply a wide pneumatic tourniquet to small pediatric patients to make a wide operation field. Finally, there was a risk of contamination of the surgical field due to the multiple reused pneumatic tourniquets. After we started using SSRTs, we did not use Esmarch bandages or pneumatic tourniquets, so I thought it was unethical to use them just for comparison.

Therefore, we added this limitation in the discussion section for the need of comparative study. (line 299).

Revision:

  1. Line 299) Even though these characteristics in pediatric studies, further study should be considered for comparison with current widely used pneumatic tourniquets or Esmarch bandages.

--------------------------------------------------------------------------------------------------------------------------

Authors reply:

For neurologic symptoms or tourniquet pain, we added information from reference articles. We greatly appreciate the thoughtful comments from the reviewer.

Revision:

  1. Line 274) According to previous studies, the incidence of pain and paresthesia among patients who underwent silicone ring tourniquet application was comparable or lower than the incidence of these complications among patients who underwent pneumatic tourniquet application, its incidence is 1/50000 and 1/5000, respectively [12,21,22]. Lee et al. recently reported in adult studies that 13.3% patients felt higher pain in the pneumatic tourniquet, 76.7% patients felt the same, and 10% patients experienced higher pain in the silicon ring tourniquet-applied lower extremity [23]. However pain associated with the use of tourniquets was first studied in 1952 and a number of mechanisms have been proposed as the cause. The exact etiology is unclear, but it is thought to be due to a cutaneous neural mechanism. The incidence of tourniquet pain was directly related to the duration of tourniquet use and was higher in cases with regional aesthesia [24]. The most important point was neurological compromise occurred in silicone ring tourniquet with over 120 minute of application time, over recommended time. You should adhere to the principle of tourniquet use, including the time of application.

Reviewer 2 Report

Bae et al  J Personalized Medicine- MDPI- 052923

The authors  evaluated sterile silicone ring tourniquet  (SSRTs) placement in pediatric patients undergoing orthopedic surgery. They reported that SSRTs effectively reduced intraoperative blood loss and facilitated wide operative fields in pediatric patients with different limb sizes. The authors concluded that SSRTs facilitated quick, safe, and effective orthopedic surgery for pediatric patients. This manuscript is clearly written, interesting and  straightforward, Comments are listed below.

(1)  Abstract: Clear statement of purpose, summary of methods, results, and conclusion.

(2)  Introduction: Nice layout of background information. It is informative that the authors compared the commonly used tourniquet with their proposed SSRTs and the need in pediatric patients.  

(3)  Methods: Assessment techniques and metrics used are  appropriate.

(4)  Results: Clear presentation and informative display of actual data  to show degree of variability in patient composition.  

        (5) Discussion-Conclusion        

a) Nice to see that the authors include the limitations of their study, and that future work needs to address these limitations.

b) Would be helpful if they clarify what is the novelty, added value of their design SSRT method  and the impact on pediatric orthopedic surgery. Overall, this manuscript offers an alternative for pediatric tourniquet design.

Author Response

The responses to the reviewers’ comments and changes are presented in blue letters

Reviewer 2.

(1)  Abstract: Clear statement of purpose, summary of methods, results, and conclusion.

(2)  Introduction: Nice layout of background information. It is informative that the authors compared the commonly used tourniquet with their proposed SSRTs and the need in pediatric patients. 

(3)  Methods: Assessment techniques and metrics used are appropriate.

(4)  Results: Clear presentation and informative display of actual data to show degree of variability in patient composition. 

(5) Discussion-Conclusion       

  1. a) Nice to see that the authors include the limitations of their study, and that future work needs to address these limitations.
  2. b) Would be helpful if they clarify what is the novelty, added value of their design SSRT method and the impact on pediatric orthopedic surgery. Overall, this manuscript offers an alternative for pediatric tourniquet design.

Authors reply:

We authors are really appreciate the reviewer's positive comments for our study. As for the reviewer's comment, we revised the strength of SSRTs in line 312.

Revision:

  1. Line 311) As opposed to concerns about pressure-focused narrow tourniquets, this study shows no pain, nerve symptoms, or skin problems in vulnerable pediatric patients. In addition, un-like conventional pneumatic tourniquets or Esmarch bandages, SSRTs apply uniform pressure to the limb during application, reducing the risk of deep vein thrombosis.

Round 2

Reviewer 1 Report

I recomend your manuscript to JPM.